# Calibrating Graph Neural Networks from a Data-centric Perspective

## ABSTRACT

Graph neural networks (GNNs) have gained popularity in modeling various complex networks, *e.g.*, social network and webpage network. Despite the promising accuracy, the confidences of GNNs are shown to be miscalibrated, indicating limited awareness of prediction uncertainty and harming the reliability of model decisions. Existing calibration methods primarily focus on improving GNN models, *e.g.*, adding regularization during training or introducing temperature scaling after training. In this paper, we argue that the miscalibration of GNNs may stem from the graph data and can be alleviated through topology modification. To support this motivation, we conduct data observations by examining the impacts of *decisive* and *homophilic* edges on calibration performance, where decisive edges play a critical role in GNN predictions and homophilic edges connect nodes of the same class. By assigning larger weights to these edges in the adjacency matrix, we observe an improvement in calibration performance without sacrificing classification accuracy. This suggests the potential of a data-centric approach for calibrating GNNs. Motivated by our observations, we propose Data-centric Graph Calibration (DCGC), which uses two edge weighting modules to adjust the input graph for GNN calibration. The first module learns the weights of decisive edges by parameterizing the adjacency matrix and enabling backpropagation of the prediction loss to edge weights. This emphasizes critical edges that fit the prediction needs. The second module computes weights for homophilic edges based on predicted label distributions, assigning larger weights to edges with stronger homophily. These modifications operate at the data level and can be easily integrated with temperature scaling-based methods for better calibration. Experimental results on 8 benchmark datasets demonstrate that DCGC achieves state-of-the-art calibration performance, with an average relative improvement of 36.4% in ECE, while maintaining or even slightly improving classification accuracy. Ablation studies and hyper-parameter analysis further validate the effectiveness and robustness of our proposed method DCGC.

## CCS CONCEPTS

• **Computing methodologies** → **Neural networks**.

## KEYWORDS

Graph Neural Network, Calibration, Data-centric Learning

**ACM Reference Format:**
Anonymous Author(s). 2018. Calibrating Graph Neural Networks from a Data-centric Perspective. In *Proceedings of Make sure to enter the correct conference title from your rights confirmation emai (Conference acronym 'XX).* ACM, New York, NY, USA, 11 pages. https://doi.org/XXXXXXX.XXXXXXX

## 1 INTRODUCTION

With the widespread applications of complex networks in various domains, the task of node classification has attracted significant attention over the last decade [22, 26, 35, 36, 39]. As a powerful framework for learning representations of graph-structured data, graph neural networks (GNNs) have demonstrated promising accuracy on various benchmarks of node classification [3, 7, 15].

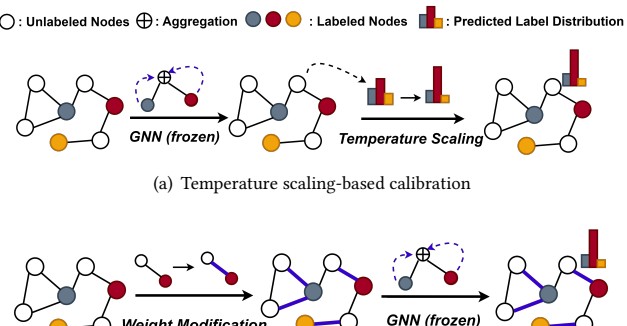

(a) Temperature scaling-based calibration

(b) Data-centric calibration

**Figure 1: Comparison between (a) previous temperature scaling-based methods [10, 33] and (b) our proposed data-centric approach. Previous work focuses on tuning temperatures in the final softmax function, while this work focuses on modifying the input graph instead.**

Besides the prediction accuracy, the awareness of prediction uncertainty is also desired for trustworthy GNNs [17]. For example, in safety-critical scenarios, GNNs are expected to know when their predictions are likely to be incorrect and accordingly alert human users. Recent advances [33] show that GNNs are usually under-confident in node classification task, *i.e.*, their prediction accuracies are higher than their confidence of being correct. To calibrate the confidence of GNNs, existing methods can be divided into two categories. The in-processing methods [29, 32] jointly train and calibrate GNNs by incorporating regularizations [32] or Bayesian modelings [29, 41]. The post-hoc methods [10, 33] are applied on well-trained GNNs for calibration and focus on adjusting the temperatures in the final softmax operation, known as temperature scaling [8]. Recent work [10] has shown that the post-hoc methods can achieve a better trade-off between accuracy and calibration

than in-processing ones. Thus, we follow the post-hoc setting and aim to calibrate well-trained GNNs in this paper.

However, existing calibration methods focus on improving GNN models, while we argue that the miscalibration of GNNs may come from the graph data and can be alleviated via topology modification. For example, we evaluate the expected calibration error (ECE) on Cora [38] and Photo [28] datasets with five different GNNs, including GCN [13], GraphSAGE [9], GAT[30], SGC[34] and TAGCN[2]. We find that the ECEs on Cora (10.25%-18.02%) are always larger than those on Photo (4.38%-8.27%), indicating that the calibration performance depends more on the datasets instead of GNN models in this case. Inspired by this phenomenon, as shown in Fig. 1, we innovatively propose to calibrate GNNs from a data-centric perspective: *can we froze the well-trained GNNs and modify the graph data instead for better calibration performance without losing accuracy?*

To support the data-centric motivation, we further conduct data observations by analyzing the impacts of *decisive* and *homophilic* edges on calibration performance. Specifically, decisive edges refer to the edges critical for the prediction of a GNN; while homophilic edges refer to the edges whose endpoints belong to the same class. By simply assigning larger weights to decisive or homophilic edges in the adjacency matrix, we find that the calibration performance can be improved without significant drop in classification accuracy, showing the potential of data-centric calibration. But note that the definitions of both decisive and homophilic edges in the above observations involves the ground truth classes of unlabeled nodes, and thus cannot be directly used in practice.

Inspired by the observations, we propose Data-centric Graph Calibration (DCGC) with two edge weighting modules to adjust the input graph. The two modules are respectively inspired by the decisive and homophilic edges, and processed sequentially: (1) For weight learning of decisive edges, we parameterize the adjacency matrix and enable the prediction loss to backpropagate to edge weights. In this way, the edge weights can automatically fit the need of label prediction, and critical edges will be emphasized. (2) For weight computation of homophilic edges, we quantify the homophily of each edge by predicted label distributions, and adaptively assign larger weights to edges with stronger homophily. Moreover, the above modifications of edge weights operate on data level, and can be easily integrated with various temperature scaling-based methods [8, 10, 33] for better calibration. Experiments on 8 benchmark datasets show that the proposed DCGC can achieve state-of-the-art (SOTA) calibration performance with 36.4% average relative improvement of ECE, and can even slightly improve the classification accuracy. Ablation studies and hyper-parameter analysis further demonstrate our effectiveness and robustness.

To summarize, our contributions are three-fold:

• We innovatively propose to calibrate GNNs from a data-centric perspective, which aims to modify the graph data for better calibration performance without losing accuracy.

• We propose a novel calibration method named DCGC by assigning larger weights to decisive and homophilic edges. The proposed DCGC operates on data level, and can be easily integrated with previous temperature scaling-based methods.

• Experiments show that DCGC can effectively calibrate different GNNs on 8 benchmark datasets, and achieves SOTA calibration performance with 36.44% average relative improvement of ECE.

## 2 RELATED WORK

### 2.1 Graph Neural Network

During the last decade, various GNN models have been proposed and show promising results in modeling structure data. As one of the most popular GNNs, GCN [13] aggregates node information from neighborhood structures using graph convolutional layers. GraphSAGE[9] enables inductive representation learning by sampling neighbor nodes and aggregating their features with multiple pooling techniques. GAT [30] incorporates attention mechanisms to learn node representations by assigning different importance weights to neighboring nodes. Simple Graph Convolution (SGC)[34] simplifies the graph convolution operation by a linear transformation on the node features. In addition to the mentioned models, there are also many GNN variants specialized for graph-level modeling, such as MoNet [20] and GIN [37]. Most GNNs focus on improving expressive power [40], and target on accuracy instead of reliability.

### 2.2 Confidence Calibration of Neural Networks

**General calibration methods.** Confidence calibration methods aim to enhance the reliability of predicted probabilities. These methods can be categorized into in-processing methods and post-hoc methods. In-processing methods, incorporate calibration directly into the training process. These methods add specific regularization terms to the loss function, encouraging the model to produce well-calibrated probabilities, such as Focal loss [16] and Maximum Mean Calibration Error (MMCE) [14]. Additional, there are some in-processing methods to tackle confidence calibration by estimating the uncertainty associated with the predictions. These techniques aim to provide not only accurate probabilities but also reliable estimates of uncertainty or confidence intervals. Bayesian neural networks [5] and Monte Carlo (MC) dropout [6] are examples that leverage probabilistic models to capture and quantify uncertainty. In contrast, post-hoc methods focus on adjusting the predicted probabilities after the initial model training. Techniques like Platt Scaling [25] and Temperature Scaling (TS) [8] fall into this category. These methods operate on the output scores of the trained model and aim to align them with the true probabilities.

**Calibration methods for GNNs.** In recent years, there has been a growing interest in developing confidence calibration methods for GNNs. These methods can also be classified into two categories as mentioned above. For in-processing methods, Graph Calibration Loss (GCL) [32] achieves calibration by adding a minimal-entropy regularizer to the KL divergence. Some approaches can calibrate GNNs by reducing model uncertainty, such as Graph-based Kernel Dirichlet distribution Estimation (GKDE) [41] and Graph Posterior Network (GPN) [29]. For post-hoc methods, CaGCN [33] introduces the confidence propagation mechanism to calibrate GNN using GCN as a topology-aware post-hoc calibration function. Essentially, CaGCN calibrates GNN by using GCN to generate node-specific temperatures. Similar to CaGCN, GATS [10] also employs a post-hoc function to obtain node-specific temperatures, and show that post-hoc methods are superior to in-processing ones.

However, existing methods focus on improving GNN models for calibration, and ignore the possibility of calibrating GNNs from the data level. To the best of our knowledge, we are the first work to calibrate GNNs from a data-centric perspective.

## 3 PRELIMINARY

**Definition 1: Semi-supervised Node Classification.** Let's consider a graph $\mathcal{G} = (\mathcal{V}, \mathcal{E}, \mathcal{X})$ with node labels $\mathcal{Y}$, where $\mathcal{V}$ represents the set of nodes, $\mathcal{E}$ represents the set of edges, and $\mathcal{X}$ represents the node features. Each node $v \in \mathcal{V}$ is associated with a feature vector $x_v \in \mathcal{X}$ and ground-truth label $y_v \in \mathcal{Y} = \{1, ..., K\}$, and each edge $(v, u) \in \mathcal{E}$ represents the relationship between nodes $v$ and $u$. Given a limited number of labeled examples as the labeled set $\mathcal{L} \subset \mathcal{V}$, the goal of semi-supervised node classification is to assign class labels to the nodes in the unlabeled set $\mathcal{U} = \mathcal{V} \setminus \mathcal{L}$.

**Definition 2: Graph Neural Network.** A graph neural network (GNN) is a parametric model that computes representations for each node by aggregating information from its neighborhood. Formally, the computation is performed by iterating over multiple layers: in each layer $l$, the representation $h_v^l$ of node $v$ will be updated by combining the representations of $v$ and its neighbors in the previous layer:

$$h_v^l = \psi^l(h_v^{l-1}, \bigoplus_{u \in \mathcal{N}(v)} \phi^l(h_v^{l-1}, h_u^{l-1})), \tag{1}$$

where $\mathcal{N}(v) = \{u | (u, v) \in \mathcal{E}\}$ is the neighbor set of node $v$, $\oplus$ denotes a differentiable, permutation-invariant function (*e.g.*, sum, mean or max), $\psi, \phi$ denote differentiable transformation functions such as multi-layer perceptrons (MLPs), and $h_v^0$ is the initial feature vector $x_v$. For the semi-supervised classification task, the representation $h_v^L$ in the final layer $L$ is usually $K$-dimensional, and can be converted into label distribution $z_v$ by the softmax function:

$$z_{v,k} = \frac{\exp(h_{v,k}^L)}{\sum_{k'=1}^K \exp(h_{v,k'}^L)}, \qquad \forall k = 1, 2 \ldots K, \tag{2}$$

where $z_{v,k}$ and $h_{v,k}^L$ are respectively the $k$-th elements of $z_v$ and $h_v^L$.

More briefly, the label predictions of a GNN can be computed as

$$Z = \text{softmax}(\text{GNN}_\Theta(A, \mathcal{X})) \in \mathbb{R}^{|\mathcal{V}| \times K}, \tag{3}$$

where $A \in \mathbb{R}^{|\mathcal{V}| \times |\mathcal{V}|}$ is the adjacency matrix, $\Theta$ denotes the set of trainable parameters, and each row of matrix $Z$ corresponds the predicted label distribution of a specific node. Then the GNN will employ a loss function (*e.g.*, cross entropy) to measure the discrepancy between prediction $Z$ and true labels on labeled set $\mathcal{L}$, and update parameters $\Theta$ via gradient descent.

**Definition 3: Calibration of GNNs and Expected Calibration Error (ECE).** Let $\hat{y}_v = \arg\max_k z_{v,k}$ be the label prediction of node $v$, and $p_v = \max_k z_{v,k}$ be the corresponding confidence. The GNN is calibrated if its prediction confidence aligns with the chance being correct. Formally, a GNN is *perfectly calibrated* [33] if

$$\Pr[\hat{y}_v = y_v | p_v = p] = p, \qquad \forall p \in [0, 1]. \tag{4}$$

For example, if the GNN makes 100 predictions with confidence 0.9, then 90 of them are correct.

One common metric to quantify the calibration of neural networks is the expected calibration error (ECE) [21], which calculates the average discrepancy between predicted probabilities and observed accuracies in different bins. Formally, we first divide unlabeled nodes into $N$ equally spaced bins based on their confidence scores, and denote the set of nodes in the $n$-th bin as $B_n$. Then we compute the gap between the accuracy and average confidence for each bin and take the weighted average of all bins as ECE:

$$\text{Acc}_n = \frac{1}{|B_n|} \sum_{v \in B_n} \mathbf{1}(y_v = \hat{y}_v), \quad \text{Conf}_n = \frac{1}{|B_n|} \sum_{v \in B_n} p_v,$$

$$\text{ECE} = \sum_{n=1}^N \frac{|B_n|}{|\mathcal{U}|} |\text{Acc}_n - \text{Conf}_n|, \tag{5}$$

where $\mathbf{1}(\cdot)$ is the indicator function. A lower ECE value indicates better calibration, *i.e.*, the model's confidence aligns well with the accuracy of its predictions. Following previous GNN calibration methods [33], we set the number of bins $N = 20$ for evaluation in practice. For brevity, we regard the computation of ECE as a function $\text{ECE}(Z)$ operating on prediction $Z$ in the following sections.

## 4 OBSERVATION

In this section, we aim to investigate the factors influencing the calibration of GNNs, and present two key observations to motivate our method design. Specifically, we respectively explore the impacts of *decisive* edges and *homophilic* edges on calibration performance.

### 4.1 Setup

We conduct observation experiments[1] on 8 datasets with GCN [13] and GraphSAGE [9]. For each dataset, we first train GNNs in a standard semi-supervised setting [10], and evaluate the calibration performance by ECE. Then we adjust the adjacency matrix by highlighting some key edges, and re-evaluate the calibration performance without updating the parameters of GNNs. Finally, we observe how the ECE varies when the adjacency matrix $A$ changes to $A'$, *i.e.*, comparing $\text{ECE}(\text{softmax}(\text{GNN}_\Theta(A, \mathcal{X})))$ with $\text{ECE}(\text{softmax}(\text{GNN}_\Theta(A', \mathcal{X})))$.

### 4.2 Impact of Decisive Edges

During the computation process of a GNN, some edges are more important than others for the final predictions. For example, in a citation network, some citation relationship may provide useful contextual information for a GNN to decide the categories of unlabeled papers. In this subsection, we aim to investigate whether the calibration performance can be improved by enlarging the weights of such decisive edges.

To quantify the importance of each edge, we can calculate the change in the test loss of the GNN model when each edge is removed from the graph. However, this calculation can be computationally expensive. Therefore, as an alternative approach, we compute the gradients of test loss $\text{Loss}_{test}$ with respect to the adjacency matrix $A$, and take the absolute value as edge importance:

$$\nabla A = |\frac{\partial \text{Loss}_{test}}{\partial A}|. \tag{6}$$

These gradients represent the sensitivity of the model classification results to the edges. Larger gradient magnitudes indicate that the corresponding edges have a stronger impact on the predictions and can be considered more decisive.

---

[1]The dataset descriptions and detailed settings will be presented in Section 6.1.

**Table 1: Calibration performance with original/modified graphs on 8 datasets. Here Modified-D and Modified-H represent the modified graphs based on decisive and homophilic edges, respectively. Decisive/homophilic edges are assigned with larger weights than unimportant/heterophilic ones. ECE scores (%) are the lower the better.**

| Model | Structure | Cora | Citeseer | Pubmed | Photo | Computers | CoraFull | Arxiv | Reddit |
|-------|-----------|------|----------|--------|-------|-----------|----------|-------|--------|
| GCN | Original | 14.43±4.52 | 14.42±4.17 | 8.41±1.29 | 7.49±1.14 | 5.92±0.29 | 14.31±0.54 | 8.00±0.15 | 5.18±0.23 |
|  | Modified-D | 14.01±3.54 | 13.97±3.24 | 7.06±1.20 | 4.29±0.56 | 4.35±0.18 | 12.84±0.41 | 7.10±0.13 | 3.45±0.19 |
|  | Modified-H | 13.61±3.92 | 14.35±3.66 | 8.29±1.01 | 6.22±1.01 | 5.07±0.51 | 13.95±0.51 | 7.70±0.12 | 2.37±0.21 |
| GraphSAGE | Original | 10.25±5.27 | 10.82±4.74 | 7.43±2.23 | 8.27±2.60 | 7.22±0.78 | 13.92±1.21 | 8.79±1.52 | 9.67±0.31 |
|  | Modified-D | 8.22±1.61 | 9.65±3.52 | 6.85±1.45 | 4.53±1.00 | 6.41±0.76 | 9.95±0.73 | 8.42±1.39 | 5.74±0.27 |
|  | Modified-H | 4.22±1.86 | 5.80±1.08 | 4.00±0.78 | 2.00±1.00 | 2.93±0.95 | 4.17±1.14 | 2.02±1.12 | 4.93±0.24 |

Next, we will modify the adjacency matrix by assigning larger weights to decisive edges. Since the gradient values may vary significantly, we take the logarithm of the gradients and divide them by the median to normalize the weight distribution, making it more comparable across different edges. Formally, the edge weight matrix is computed as:

$$\nabla A_{weight} = \frac{\log(\nabla A + \sigma)}{\text{median}(\log(\nabla A + \sigma))}, \qquad (7)$$

where $\sigma$ is a constant slightly greater than 1. Then we modify the adjacency matrix as $A' = A \odot \nabla A_{weight}$ and re-evaluate the calibration performance as $\text{ECE}(\text{softmax}(\text{GNN}_\Theta(A', \mathcal{X})))$, where $\odot$ is the element-wise product.

Table 1 shows the calibration performance with original/modified graphs on 8 datasets over 10 runs. The results show that enlarging the weights of decisive edges has a positive impact on calibration performance for both GCN and GraphSAGE over all 8 datasets.

### 4.3 Impact of Homophilic Edges

Note that the above decisive edges are model-specific. For the node classification task, the edges whose endpoints belong to the same category are also critical for message passing and irrelevant to GNN models. In this subsection, we aim to investigate whether the calibration performance can be improved by enlarging the weights of such homophilic edges.

In this observation, we use ground truth labels to define homophilic edges and set edge weights heuristically. Specifically, the weights of homophilic edges are twice as the weights of heterophilic ones. Similar to the previous subsection, we then modify the adjacency matrix accordingly and re-evaluate the calibration performance.

Table 1 shows the calibration performance with original/modified graphs on 8 datasets over 10 runs. We can see that enlarging the weights of homophilic edges also brings a notable improvement for calibration performance.

### 4.4 Discussion

It is worth noting that decisive edges and homophilic edges are quite independent with each other. For instance, given a random homophilic edge and a random heterophilic one, the probability that the homophilic edge has a larger importance score as Eq. (6) is around 0.55 (0.5 if they are fully independent). Therefore, the above two observations do not overlap.

More importantly, the above modifications of adjacency matrices will not bring significant drop in classification accuracy when improving the calibration performance. Therefore, these observations show the potential that we can calibrate a well-trained GNN by modifying edge weights without changing GNN parameters.

However, the weight computation of both decisive and homophilic edges involves the ground truth classes of unlabeled nodes, and thus cannot be directly used in practice. Hence, in next section, we will propose our method that can identify these two kinds of edges and enlarge their weights without using ground truth labels.

## 5 METHODOLOGY

In this section, we propose a data-centric approach for calibrating GNNs, named Data-Centric Graph Calibration (DCGC). DCGC can be applied on any well-trained GNNs, and is also compatible with previous temperature scaling-based methods [8, 10, 33].

### 5.1 Overview

The overview of our method is shown in Figure 2. Given a well-trained GNN, we design two modules to adjust the edge weights of the input graph. The two modules are respectively inspired by the observations of decisive and homophilic edges, and processed sequentially in our method. Afterward, temperature scaling-based methods [8, 10, 33] can be integrated to adjust the sharpness of label predictions. Note that the parameters of learned GNNs are frozen in the entire pipeline.

### 5.2 Weight Learning of Decisive Edges

Inspired by the first observation, we propose to parameterize the adjacency matrix, and enable the prediction loss to backpropagate to edge weights. In this way, the edge weights can be automatically adjusted to fit the need of label prediction and assign larger weights for critical edges.

Specifically, we first design a trainable module that estimates the weight of an edge based on the representations of its endpoints. Formally, for each edge $(v, u) \in \mathcal{E}$, we first encode nodes $v$ and $u$ as $h_v$ and $h_u$ by the well-trained GNN, and then introduce a 2-layer MLP for weight computation:

$$w_{v,u}^1 = \max(\text{MLP}_\Omega(\text{concat}(h_v, h_u)), 0), \qquad (8)$$

where $\Omega$ denotes the set of parameters in the MLP, and $\text{concat}(\cdot, \cdot)$ is the concatenation operation. We gather the weights of all edges

**Figure 2: Illustration of our proposed Data-Centric Graph Calibration (DCGC). The entire pipeline of calibrating GNNs with DCGC is as follows: (a) train the GNN to be calibrated in a standard semi-supervised manner; (b) learn larger weights for decisive edges and accordingly modify the adjacency matrix; (c) assign larger weights for homophilic edges and accordingly modify the adjacency matrix; (d) integrate DCGC with any temperature scaling-based method.**

and denote the weight matrix as $W^1$, a sparse matrix with the same shape as the adjacency matrix. Then we re-calculate the predictions of the GNN by the modified adjacency matrix as $Z^1 = \text{softmax}(\text{GNN}_\Theta(A \odot W^1, \mathcal{X}))$.

To optimize this edge weighting module, we simply minimize the cross-entropy loss on the validation set[2] by gradient descent:

$$\min_\Omega - \sum_{v \in \mathcal{D}} \sum_{k=1}^K \mathbf{1}(y_v = k) \log z_{v,k}^1, \qquad (9)$$

where $\mathcal{D}$ is the validation set, and $z_{v,k}^1$ is the corresponding element in matrix $Z^1$.

Compared with the observation of decisive edges in Section 4.2, we use the prediction loss on validation set instead, and dynamically adjust the edge weights by optimization. By minimizing Eq. (9), the module learns to assign larger weights to edges critical to the classification task, and help improve the calibration performance.

### 5.3 Weight Computation of Homophilic Edges

Inspired by the second observation, we propose to quantify the homophily of each edge by predicted label distributions, and assign larger weights to edges with stronger homophily. This module requires no training process, and is heuristically designed.

Specifically, we first measure the homophily of edge $(v, u)$ by the Euclidean distance between the label predictions $z_v^1$ and $z_u^1$, and then define the edge weight as

$$w_{v,u}^2 = \left( \| \text{softmax-TS}(z_v^1, \beta) - \text{softmax-TS}(z_u^1, \beta) \|_2 + \alpha \right)^{-1}, \quad (10)$$

where $\text{softmax-TS}(\cdot, \cdot)$ is the softmax function with temperature scaling:

$$\text{softmax}(z_v, \tau) = \frac{\exp(z_v/\tau)}{\sum_{k'=1}^K \exp(z_{v,k'}/\tau)}. \qquad (11)$$

Here temperature $\tau > 0$ controls the sharpness of predicted label distribution: a smaller $\tau$ will push the prediction towards one-hot vector. Besides $\beta$, another hyper-parameter $\alpha$ ensures that the edge

---

[2]Previous GNN calibration methods [10, 33] also use the validation set for parameter learning.

weight will not be excessively large when the predicted distributions are too close to each other.

Then we gather the weights of all edges and denote the weight matrix as $W^2$, which is also a sparse matrix as $W^1$. The label predictions can be calculated as $Z^2 = \text{softmax}(\text{GNN}_\Theta(A \odot W^1 \odot W^2, \mathcal{X}))$.

Compared with the observation of homophilic edges in Section 4.3, we use the predicted label distributions to define homophilic edges instead, and design a smooth function as in Eq. (10) to compute edge weights. The module heuristically assigns larger weights to edges with similar labels, and help improve the calibration performance.

---

**Algorithm 1** Data-Centric Graph Calibration (DCGC)

---

**Require:** Graph $\mathcal{G} = (\mathcal{V}, \mathcal{E}, \mathcal{X})$, well-trained GNN model $\text{GNN}_\Theta$, initialized parameter $\Omega$, hyper-parameters $\alpha, \beta$;

**Ensure:** Learned parameter $\Omega$ and prediction $Z^2$;

1: Encode every node $v$ as $h_v$ by $\text{GNN}_\Theta(A, \mathcal{X})$;
2: **while** not converge **do**
3:     Compute each element $w_{v,u}^1$ of edge weight matrix $W^1$ with parameter $\Omega$ as Eq. (8);
4:     Compute label prediction $Z^1 = \text{softmax}(\text{GNN}_\Theta(A \odot W^1, \mathcal{X}))$;

5:     Backpropagate $\Omega$ by minimizing the loss as Eq. (9);
6: **end while**
7: Compute each element $w_{v,u}^2$ of edge weight matrix $W^2$ with prediction $Z^1$ as Eq. (10);
8: Encode every node $v$ as $h_v^2$ by $\text{GNN}_\Theta(A \odot W^1 \odot W^2, \mathcal{X})$;
9: Employ any temperature scaling-based methods to assign temperature $\tau_v$ for each node $v$;
10: Compute the final label distribution prediction $z_v^2$ for each node $v$ as $z_v^2 = \text{softmax}(h_v^2, \tau_v)$;

---

### 5.4 Integration with Temperature Scaling

The above modifications of edge weights operate on data level, and can be easily integrated with previous temperature scaling-based methods [8, 10, 33] for better calibration.

Specifically, given the weight matrices $W^1, W^2$, we denote the representation of node $v$ encoded by $\text{GNN}_\Theta(A \odot W^1 \odot W^2, X)$ as $h_v^2$. Then we rewrite the computation of prediction $Z^2$ by defining the prediction of node $v$ as $z_v^2 = \text{softmax}(h_v^2, \tau_v)$, where $\tau_v$ is the temperature. In practice, the temperatures can be global [8], class-specific [8] or node-specific [10, 33]. We will evaluate the integration with different temperature scaling methods in our experiments.

We present the pseudo code of DCGC in Algorithm 1. Finally, we take $Z^2$ as algorithm output and compute $\text{ECE}(Z^2)$ for evaluation. Note that we only model edge weights for existing edges, and thus enjoy linear computational complexity. We will demonstrate the efficiency of our algorithm in Section 6.4.

## 6 EXPERIMENT

In this section, we conduct experimental evaluation to answer the following research questions (RQs): **RQ1:** How does DCGC perform compared with state-of-the-art GNN calibration algorithms? **RQ2:** How does the two weighting modules contribute to the overall performance? **RQ3:** How about the efficiency of DCGC? **RQ4:** How about the hyper-parameter sensitivity of DCGC?

### 6.1 Experimental Settings

*6.1.1 Dataset.* We consider 8 popular graph datasets from various domains to evaluate the effectiveness of our proposed DCGC: Cora [38], Citeseer [38], Pubmed [38], CoraFull [1], Photo [28], Computers [28], Arxiv [11] and Reddit [9]. Cora, Citeseer, Pubmed are three well-known citation network datasets and CoraFull is the extended version of Cora. Photo and Computers are subsets of the Amazon co-purchase graph dataset, where nodes represent different Amazon goods and edges indicate frequent co-purchases between goods. Arxiv dataset comes from OGB [11], and is an arxiv citation network extracted from the Microsoft Academic Graph [31]. Reddit dataset is a large-scale graph where nodes represent posts, and edges represent that the same user comments on both. Statistics of these datasets are listed in Appendix A. Following the partition settings of GATS[10], we use 10% random nodes as the training set, 5% random nodes as the validation set, and the other 85% nodes as the test set for all 8 datasets.

*6.1.2 GNNs to be Calibrated.* In our experiments, we train GCN [13] and GraphSAGE [9] on each dataset as the GNNs to be calibrated. For GCN, we use a 2-layer architecture and 16-dimensional hidden size. For GraphSAGE, we adopt the default average aggregation variant with 2-layer architecture and 16-dimensional hidden size. Model parameters of GCN and GraphSAGE are fixed after training.

*6.1.3 Baselines.* To prove the effectiveness and compatibility of DCGC, we consider four temperature scaling-based baselines for comparison and integration: Temperature Scaling (TS) [8], Vector Scaling (VS) [8], CaGCN [33] and Graph Attention Temperature Scaling (GATS) [10]. TS and VS are calibration methods designed for general classification task, while CaGCN and GATS are specialized for GNNs. Specifically, TS employs a global temperature parameter in the final softmax function. VS learns class-specific temperatures for calibration. CaGCN calibrates GNNs by employing GCN to generate node-specific temperatures. Similar to CaGCN, GATS also learns node-specific temperatures based on a heuristic formula. We integrate our DCGC with each of the four scaling methods, respectively. All calibration methods are trained on the validation set, including our DCGC. The implementation of GATS requires the entire graph, and thus is not compatible with sampling or batch processing of large-scale graphs. Hence we ignore the results of GATS on Reddit dataset.

*6.1.4 Ablated Variants.* We consider two ablated variants of DCGC for comparison: $\text{DCGC}_{\text{w/o D}}$ indicates that the weighting module of decisive edges is removed; while $\text{DCGC}_{\text{w/o H}}$ indicates that the weighting module of homophilic edges is removed.

*6.1.5 Evaluation Metrics.* Following previous calibration methods [33], we adopt Expected Calibration Error (ECE) with 20 bins as the metric for calibration. Besides, since the modification of graph structure may influence the predicted label $\hat{y}_v$, we also adopt classification accuracy as another evaluation metric. In addition, we train GNNs with five random dataset divisions, and conduct five runs of calibration for each well-trained GNN, *i.e.*, a total of 25 runs for each pair of dataset and GNN. We report both the average and standard deviation.

*6.1.6 Implementation Details.* For hyper-parameters in DCGC, we tune $\alpha \in \{0.1, 0.3, 0.5, 1\}$ and $\beta \in \{0.1, 0.2, 1, 10\}$ based on the performance on validation set. For the MLP in the decisive edge weighting module, we adopt 2-layer architecture and 2× input dimension as hidden size, *i.e.*, 4× the number of classes. We fix weight decay as 0.005 and learning rate as 0.01, and train calibration methods for at most 1000 epochs using early stop of 200 patience.

### 6.2 Main Results (RQ1)

The results of calibration and accuracy are shown in Table 2 and 3, respectively, and we have the following observations:

(1) Before evaluating the calibration performance, we first show that our data-centric approach will not harm the classification accuracy. As shown in Table 2, compared with uncalibrated GCN and GraphSAGE, DCGC and its variants have competitive or even better prediction accuracy. The average relative improvement of DCGC over GCN/GraphSAGE are 0.51%/0.32%, respectively. Thus, instead of sacrificing precision as a trade-off, our proposed DCGC can bring accuracy gains as a by-product.

(2) By integrating with temperature scaling-based methods, DCGC can achieve SOTA calibration performance on all datasets and GNNs. In terms of ECE, the average relative improvement over TS, VS, CaGCN and GATS are respectively 39.8%, 38.6%, 34.4% and 33.0%. In fact, even when equipped with the vanilla TS, DCGC+TS is already better than SOTA GNN calibration methods, *i.e.,* CaGCN and GATS. This observation validates the effectiveness and compatibility of our proposed data-centric calibration.

(3) For some datasets such as Pubmed, calibration methods specialized for GNNs (CaGCN and GATS) do not show significant advantages over graph-irrelevant ones (TS and VS). Thus, only utilizing graph information in modeling node-specific temperatures may not make full use of the structure data. In contrast, our DCGC calibrates GNN predictions by modifying the input graph, and thus has stronger impact on the calibration performance.

**Table 2: The classification accuracy (in percentage) of uncalibrated GNNs and their calibrated versions based on DCGC. Note that temperature scaling tricks will not affect the accuracy. The accuracy is the higher the better.**

| Model | Method | Cora | Citeseer | Pubmed | Photo | Computers | CoraFull | Arxiv | Reddit |
|---|---|---|---|---|---|---|---|---|---|
| GCN | Original | 82.84±1.64 | 72.40±1.14 | 87.23±0.24 | 92.50±0.55 | 87.88±0.45 | 62.04±0.58 | 63.94±0.57 | 90.34±0.38 |
| | DCGC$_{w/o\ D}$ | 82.80±1.73 | 72.38±0.81 | 87.59±0.38 | 92.55±0.46 | 87.43±0.33 | 61.97±0.60 | 64.14±0.25 | 90.41±0.33 |
| | DCGC$_{w/o\ H}$ | 82.73±1.54 | 72.72±1.26 | 87.63±0.40 | 93.34±0.29 | 89.07±0.53 | 63.12±0.58 | 64.05±0.16 | 90.36±0.27 |
| | DCGC | 82.62±1.72 | 72.40±0.94 | 87.72±0.48 | 93.39±0.38 | 88.80±0.42 | 63.09±0.60 | 64.07±0.12 | 90.37±0.27 |
| GraphSAGE | Original | 83.49±0.57 | 70.93±0.68 | 86.58±0.36 | 92.29±0.39 | 87.09±0.83 | 60.02±0.25 | 61.79±0.35 | 90.21±0.12 |
| | DCGC$_{w/o\ D}$ | 83.65±0.72 | 71.09±0.74 | 86.85±0.47 | 92.30±0.42 | 87.11±0.87 | 60.05±0.25 | 61.66±0.23 | 90.29±0.16 |
| | DCGC$_{w/o\ H}$ | 83.66±0.50 | 71.89±0.52 | 86.98±0.54 | 92.43±0.24 | 87.70±0.76 | 60.32±0.54 | 61.50±0.20 | 90.39±0.08 |
| | DCGC | 83.78±0.60 | 71.04±0.61 | 87.24±0.69 | 92.51±0.31 | 87.74±0.76 | 60.28±0.24 | 61.46±0.17 | 90.40±0.09 |

**Table 3: The calibration results of different methods on eight datasets. ECE scores (%) are the lower the better.**

| Model | Method | Cora | Citeseer | Pubmed | Photo | Computers | CoraFull | Arxiv | Reddit |
|---|---|---|---|---|---|---|---|---|---|
| GCN | Original | 14.43±4.52 | 14.42±4.17 | 8.41±1.29 | 7.49±1.14 | 5.92±0.29 | 14.31±0.54 | 8.00±0.15 | 5.18±0.23 |
| | TS | 6.60±1.83 | 10.22±1.92 | 4.43±0.58 | 3.16±1.02 | 3.92±1.56 | 11.00±0.78 | 6.39±0.31 | 5.12±0.22 |
| | DCGC+TS | 4.89±1.41 | 8.13±2.36 | 2.18±0.71 | 1.72±0.62 | 1.93±0.50 | 5.63±0.78 | 4.26±0.37 | 4.17±0.32 |
| | VS | 8.26±1.80 | 10.86±1.38 | 5.02±0.68 | 4.54±0.96 | 4.46±1.31 | 13.68±0.37 | 7.68±0.21 | 4.36±0.05 |
| | DCGC+VS | 6.04±1.67 | 8.86±1.69 | 2.50±0.85 | 1.77±0.49 | 1.67±0.70 | 8.32±0.85 | 4.60±0.27 | 3.84±0.27 |
| | CaGCN | 6.88±1.29 | 8.41±1.87 | 3.52±0.56 | 1.75±0.72 | 2.94±3.33 | 7.09±0.58 | 3.87±0.39 | 2.92±0.14 |
| | DCGC+CaGCN | 5.42±1.25 | 6.68±1.85 | 1.68±0.54 | 1.11±0.24 | 2.55±2.84 | 4.52±0.47 | 2.86±0.37 | 1.23±0.26 |
| | GATS | 5.27±1.86 | 9.09±2.03 | 3.69±0.51 | 1.41±0.41 | 1.61±0.85 | 9.07±0.61 | 4.42±0.31 | - |
| | DCGC+GATS | 4.23±1.24 | 7.17±2.30 | 1.66±0.47 | 1.30±0.26 | 1.58±0.41 | 4.21±0.56 | 3.87±0.33 | - |
| GraphSAGE | Original | 10.25±5.27 | 10.82±4.74 | 7.43±2.23 | 8.27±2.60 | 7.22±0.78 | 13.92±1.21 | 8.79±1.52 | 9.67±0.31 |
| | TS | 9.68±3.83 | 9.42±1.68 | 5.15±0.80 | 2.76±0.79 | 2.85±0.69 | 10.54±1.33 | 7.77±0.99 | 9.05±0.20 |
| | DCGC+TS | 6.03±1.19 | 5.00±0.68 | 3.54±1.06 | 1.45±0.50 | 2.26±0.66 | 5.39±1.25 | 4.14±1.21 | 4.04±0.47 |
| | VS | 9.91±3.75 | 9.18±3.19 | 5.14±0.35 | 4.11±0.89 | 4.25±0.68 | 14.47±1.66 | 8.55±1.18 | 9.87±0.26 |
| | DCGC+VS | 5.14±0.72 | 5.91±0.76 | 2.19±0.63 | 1.62±0.71 | 2.14±0.55 | 8.28±1.63 | 5.10±1.36 | 8.16±0.36 |
| | CaGCN | 9.49±2.29 | 8.67±1.64 | 4.63±1.74 | 2.05±0.63 | 2.38±0.36 | 6.91±1.35 | 4.13±1.22 | 5.02±0.22 |
| | DCGC+CaGCN | 5.26±1.35 | 5.38±3.10 | 2.30±0.69 | 1.31±0.36 | 2.13±0.43 | 4.29±0.84 | 3.83±1.15 | 2.15±0.17 |
| | GATS | 9.68±3.38 | 8.86±2.05 | 5.04±1.33 | 2.44±0.77 | 2.76±0.58 | 8.69±1.27 | 5.96±1.21 | - |
| | DCGC+GATS | 6.99±1.61 | 6.18±1.73 | 3.70±1.25 | 1.43±0.40 | 2.31±0.67 | 4.50±0.99 | 2.92±1.16 | - |

## 6.3 Ablation Study (RQ2)

In this subsection, we will conduct ablation study to discuss the impact of two weighting modules in DCGC. The two ablated variants, *i.e.,* DCGC$_{w/o\ D}$ and DCGC$_{w/o\ H}$, respectively remove the modeling of decisive and homophilic edges. The accuracy and calibration performance of ablated models are shown in Table 2 and 4. From the results, we can see that:

(1) The full model DCGC has better calibration performance than the two ablated variants with 22.03% and 15.29% average relative improvement of ECE. Hence, both weighting modules of decisive and homophilic edges contribute to the final calibration performance of DCGC, which validates the effectiveness of our design.

(2) Compared with uncalibrated GNNs, both DCGC$_{w/o\ D}$ and DCGC$_{w/o\ H}$ have competitive or even better classification accuracy. The two ablated variants can also improve the calibration performance over temperature scaling baselines. Therefore, the two weighting modules can be deployed separately for calibrating GNNs without sacrificing prediction accuracy.

(3) The decisive edge weighting module is more important the homophilic one. Compared with DCGC$_{w/o\ D}$, DCGC$_{w/o\ H}$ has 0.44%

and 7.36% relative improvement in accuracy and calibration. A possible reason is that the decisive edge weighting module is learned via optimization instead of heuristically designed. We will explore more powerful edge weighting methods for homophilic edges in the future work.

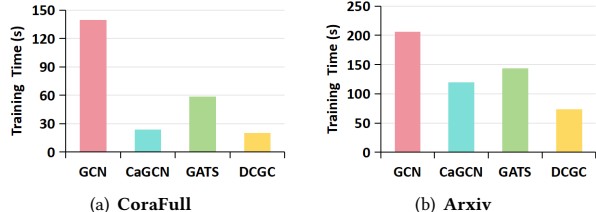

(a) **CoraFull**                    (b) **Arxiv**

**Figure 3: Training time comparison between different methods on CoraFull and Arxiv datasets.**

## 6.4 Efficiency Analysis (RQ3)

To demonstrate the efficiency of DCGC, we present the training time of DCGC and other methods on CoraFull and Arxiv datasets. We

**Table 4: The calibration results of different model variants on eight datasets. ECE scores are the lower the better.**

| Model | Method | Cora | Citeseer | Pubmed | Photo | Computers | CoraFull | Arxiv | Reddit |
|-------|--------|------|----------|--------|-------|-----------|----------|-------|--------|
| GCN | DCGC$_{w/o\ D}$+TS | 5.27±1.53 | 9.90±1.95 | 3.49±0.58 | 2.11±0.61 | 2.49±0.68 | 9.70±0.52 | 5.81±0.29 | 4.94±0.36 |
| | DCGC$_{w/o\ H}$+TS | 5.26±1.78 | 8.32±2.35 | 2.47±0.99 | 1.76±0.44 | 2.05±0.88 | 6.85±1.14 | 5.49±0.36 | 4.67±0.47 |
| | DCGC+TS | 4.89±1.41 | 8.13±2.36 | 2.18±0.71 | 1.72±0.62 | 1.93±0.50 | 5.63±0.78 | 4.26±0.27 | 4.17±0.32 |
| | DCGC$_{w/o\ D}$+VS | 7.32±1.59 | 10.50±1.36 | 4.16±0.60 | 2.62±0.79 | 2.36±0.59 | 12.51±0.49 | 6.51±0.37 | 4.21±0.14 |
| | DCGC$_{w/o\ H}$+VS | 6.61±1.89 | 9.16±1.49 | 2.95±1.09 | 1.84±0.55 | 2.03±0.96 | 9.57±0.72 | 6.04±0.33 | 3.92±0.28 |
| | DCGC+VS | 6.04±1.67 | 8.86±1.69 | 2.50±0.85 | 1.77±0.49 | 1.67±0.70 | 8.32±0.85 | 4.60±0.27 | 3.84±0.27 |
| | DCGC$_{w/o\ D}$+CaGCN | 6.00±1.34 | 8.26±1.70 | 2.48±0.51 | 1.31±0.25 | 2.55±3.18 | 6.72±0.41 | 3.42±0.23 | 2.53±0.34 |
| | DCGC$_{w/o\ H}$+GaGCN | 5.72±1.70 | 6.77±2.05 | 2.08±0.87 | 1.16±0.25 | 2.62±2.61 | 4.68±0.82 | 3.28±0.29 | 1.86±0.25 |
| | DCGC+CaGCN | 5.42±1.25 | 6.68±1.85 | 1.68±0.54 | 1.11±0.24 | 2.55±2.84 | 4.52±0.47 | 2.86±0.27 | 1.23±0.26 |
| | DCGC$_{w/o\ D}$+GATS | 4.14±1.43 | 8.64±2.25 | 2.47±0.34 | 1.43±0.11 | 1.66±0.25 | 7.90±0.43 | 4.32±0.19 | - |
| | DCGC$_{w/o\ H}$+GATS | 4.62±1.32 | 7.38±2.42 | 2.05±0.89 | 1.36±0.47 | 1.72±0.62 | 5.00±0.92 | 3.99±0.29 | - |
| | DCGC+GATS | 4.23±1.24 | 7.17±2.30 | 1.66±0.47 | 1.30±0.26 | 1.58±0.41 | 4.21±0.56 | 3.87±0.33 | - |
| GraphSAGE | DCGC$_{w/o\ D}$+TS | 6.29±0.81 | 6.98±2.35 | 3.68±1.25 | 2.73±0.75 | 2.84±0.68 | 7.23±1.38 | 7.76±0.99 | 5.05±0.24 |
| | DCGC$_{w/o\ H}$+TS | 8.20±3.90 | 8.14±2.06 | 4.71±1.60 | 1.50±0.48 | 2.33±0.66 | 8.01±2.74 | 4.10±1.18 | 4.18±0.32 |
| | DCGC+TS | 6.03±1.19 | 5.00±0.68 | 3.54±1.06 | 1.45±0.50 | 2.26±0.66 | 5.39±1.25 | 4.14±1.21 | 4.04±0.47 |
| | DCGC$_{w/o\ D}$+VS | 5.82±0.87 | 6.11±3.37 | 2.29±0.14 | 4.06±0.83 | 4.19±0.64 | 8.50±1.03 | 8.51±1.17 | 9.10±0.22 |
| | DCGC$_{w/o\ H}$+VS | 8.87±3.78 | 8.82±3.12 | 4.70±1.82 | 1.69±0.65 | 2.19±0.55 | 9.73±2.35 | 5.12±1.35 | 8.85±0.28 |
| | DCGC+VS | 5.14±0.72 | 5.91±0.76 | 2.19±0.63 | 1.62±0.71 | 2.14±0.55 | 8.28±1.63 | 5.10±1.36 | 8.16±0.36 |
| | DCGC$_{w/o\ D}$+CaGCN | 5.46±0.97 | 6.74±1.89 | 2.33±0.69 | 2.01±0.63 | 2.37±0.39 | 6.00±2.45 | 4.00±1.26 | 4.85±0.25 |
| | DCGC$_{w/o\ H}$+GaGCN | 6.65±2.10 | 6.91±1.99 | 4.16±1.60 | 1.36±0.38 | 2.18±0.49 | 7.17±1.67 | 3.89±1.17 | 2.89±0.27 |
| | DCGC+CaGCN | 5.26±1.35 | 5.38±3.10 | 2.30±0.69 | 1.31±0.36 | 2.13±0.43 | 4.29±0.84 | 3.83±1.15 | 2.15±0.17 |
| | DCGC$_{w/o\ D}$+GATS | 6.14±0.91 | 6.01±1.35 | 3.90±1.26 | 2.16±0.71 | 2.58±0.45 | 6.78±1.51 | 6.00±1.20 | - |
| | DCGC$_{w/o\ H}$+GATS | 7.55±3.20 | 7.05±1.05 | 4.39±1.55 | 1.40±0.37 | 2.39±0.68 | 7.52±2.55 | 2.86±1.12 | - |
| | DCGC+GATS | 6.99±1.61 | 6.18±1.73 | 3.70±1.25 | 1.43±0.40 | 2.31±0.67 | 4.50±0.99 | 2.92±1.16 | - |

can see that compared with the semi-supervised training of GCN, all three calibration methods are very efficient. In particular, the training time of DCGC (excluding the integration of temperature scaling) is the least. Thus, when equipping DCGC with temperature scaling-based methods for calibration, the computational overhead brought by DCGC is very limited, making DCGC applicable for large-scale graphs.

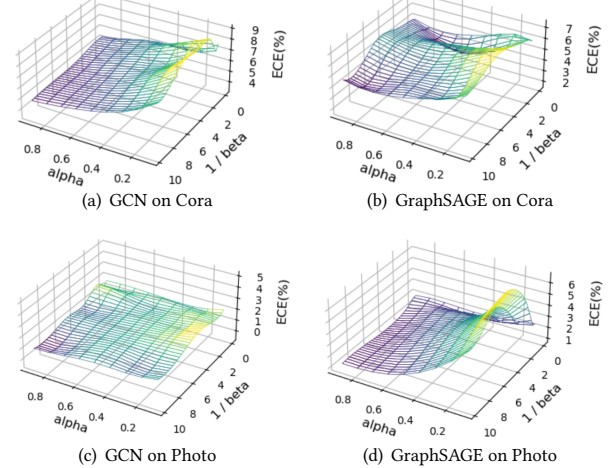

(a) GCN on Cora      (b) GraphSAGE on Cora

(c) GCN on Photo      (d) GraphSAGE on Photo

**Figure 4: Hyper-parameter sensitivity of $\alpha$ and $\beta$ in Eq. (10) on Cora and Photo. For legibility, we use $1/\beta$ as the axis.**

## 6.5 Hyper-parameter Sensitivity (RQ4)

In this subsection, we explore the influence of hyper-parameters on calibration performance. Fixed hyper-parameters of DCGC have been introduced in Section 6.1.6. Besides, DCGC has two key hyper-parameters in computing the weights of homophilic edges: $\alpha \in \{0.1, 0.3, 0.5, 1\}$ and $\beta \in \{0.1, 0.2, 1, 10\}$. Fig. 4 presents the calibration performance of DCGC under different hyper-parameters $\alpha$ and $\beta$ on Cora and Photo datasets. We can see that a larger $\alpha$ is preferred in all four cases, and the calibration performance is not very sensitive to the change of $\beta$ at this time. This suggests the robustness of DCGC under proper hyper-parameter settings.

## 7 CONCLUSION

In this paper, we propose a novel data-centric perspective for the calibration of graph neural networks, which aims to modify the graph structure for better calibration performance without sacrificing prediction accuracy. By analyzing the impact of decisive and homophilic edges on calibration, we design DCGC with two corresponding edge weighting modules that can adaptively assign larger weights to such important edges. The proposed DCGC is also highly compatible with existing temperature scaling-based methods. Experimental results on eight datasets demonstrate the effectiveness and efficiency of DCGC.

For future work, we will explore the potential of DCGC on graph-level tasks as well as other trustworthy graph learning scenarios. It is also possible to replace the heuristic design of homophilic edge weighting module with more powerful learning algorithms.

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

**Table 5: Dataset statistics of eight graph datasets.**

|  | Cora | Citeseer | Pubmed | Photo | Computers | CoraFull | Arxiv | Reddit |
|---|---|---|---|---|---|---|---|---|
| # Nodes | 2,708 | 3,327 | 19,717 | 7,650 | 13,752 | 19,793 | 169,343 | 232,965 |
| # Edges | 10,556 | 9,104 | 88,648 | 238,162 | 491,722 | 126,842 | 1,166,243 | 114,615,892 |
| # Features | 1,433 | 3,703 | 500 | 745 | 767 | 8,710 | 128 | 602 |
| # Classes | 7 | 6 | 3 | 8 | 10 | 70 | 40 | 41 |

## A    DATASET STATISTICS

We summarize the dataset statistics in Table 5 and the detailed dataset descriptions are as follows:

- **Cora, Citeseer and Pubmed:** These datasets are constructed by three text classification datasets [27]. Each node feature vector is the bag-of-words representation of a document, and each edge indicates a citation relationship between two documents. The task is to classify each document into the correct class.
- **Photo and Computers:** Both datasets are parts of the Amazon co-purchase graph [18]. Each node feature vector is the bag-of-words encoded product reviews, and each edge indicates a co-purchase relationship between two goods.
- **CoraFull:** Cora dataset is constructed on a small subset of the original citation dataset [19], while CoraFull additionally extracts the entire network.
- **Arxiv:** This dataset comes from OGB [11], and is an arxiv citation network extracted from the Microsoft Academic Graph [31]. Each node can be mapped to a research paper.
- **Reddit:** Reddit is a large online discussion forum where users can post and comment in different communities. Each node feature vector is 300-dimensional GloVe word vectors [24] of a post, and each edge indicates that the same user comments on both posts. The task is to classify which community different Reddit posts belong to.

## B    MORE EXPERIMENTAL SETTING DETAILS

### B.1    Optimizer

We choose Adam optimizer [12] with fixed learning rate as 0.01 to train GNNs and calibration models. For GNNs, we follow PyTorch Geometric (PyG) [4] by setting weight decay as 0.005 on the first layer and 0 on the second layer. For calibration methods, we fix weight decay as 0.005.

### B.2    Implementation Details

We implement GNNs and calibration methods based on PyTorch [23] and PyTorch Geometric (PyG) [4]. For all experiments, we employ GeForce RTX 2080 as our GPU device.

## C    ADDITIONAL EXPERIMENTS

Here we present extra figures of the efficiency analysis (RQ3) and hyper-parameter analysis (RQ4). These figures show similar patterns with those in Section 6.

### C.1    Training Time on Other Datasets

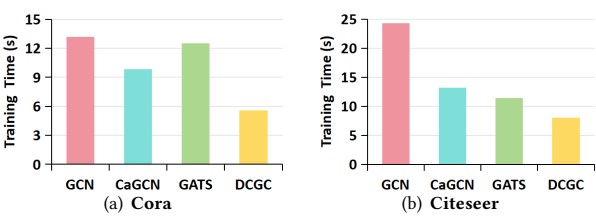

(a) **Cora**          (b) **Citeseer**

**Figure 5: Training time on Cora and Citeseer datasets.**

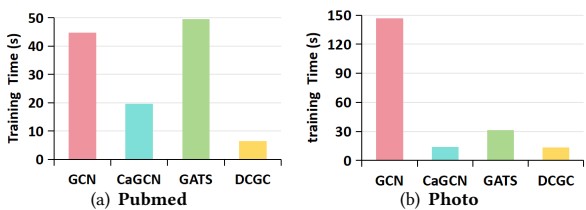

(a) **Pubmed**          (b) **Photo**

**Figure 6: Training time on Pubmed and Photo datasets.**

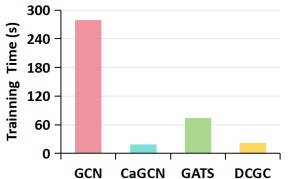

**Figure 7: Training time on Computers dataset.**

### C.2    Hyper-parameter Sensitivity on Other Datasets with GCN

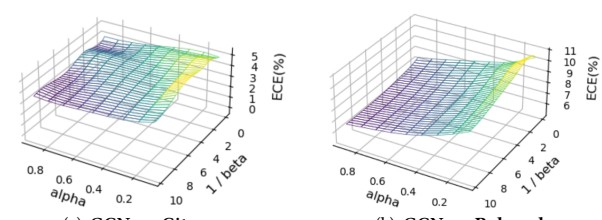

(a) **GCN on Citeseer**          (b) **GCN on Pubmed**

**Figure 8: Hyper-parameter sensitivity of $\alpha$ and $\beta$ on Citeseer and Pubmed datasets.**

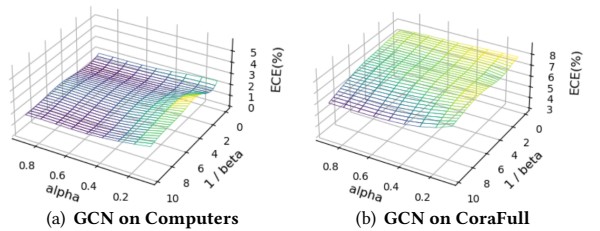

(a) GCN on Computers      (b) GCN on CoraFull

Figure 9: Hyper-parameter sensitivity of $\alpha$ and $\beta$ on Computers and CoraFull datasets.

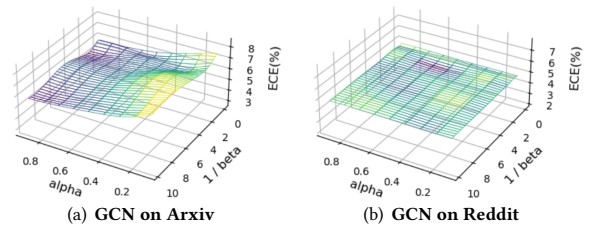

(a) GCN on Arxiv      (b) GCN on Reddit

Figure 10: Hyper-parameter sensitivity of $\alpha$ and $\beta$ on Arxiv and Reddit datasets.

Received 20 February 2007; revised 12 March 2009; accepted 5 June 2009

