# OpenReview forum: "Calibrating Graph Neural Networks from a Data-centric Perspective"
_ACM.org/TheWebConf/2024/Conference — TheWebConf24_

### Official Review · Reviewer_zSCm · 2023-11-18

**Novelty:** 5
**Technical Quality:** 5

**Review:**

The authors present an innovative data-centric approach for calibrating GNNs, addressing a critical issue in the field. By focusing on the graph data itself, particularly through the adjustment of decisive and homophilic edges, the paper demonstrates an improvement in calibration performance without sacrificing classification accuracy.

Pros:

- The research problem is significant and challenging.
- The analysis is sufficient and the key motivation is well addressed.
- Extensive experiments are conducted to evaluate the performance of the proposed method.

Cons:

- This method is an improvement on the existing framework, and the contribution seems to be insufficient.
- This paper investigates the effectiveness of GCN and GraphSAGE, and it is recommended that the author add more backbone models for evaluation, such as GAT and GIN.
- The changes in accuracy after applying DCGC are different between GraphSAGE and GCN. Would the author try to explain it?

**Questions:**

Please address my aforementioned concerns.

**Reviewer Confidence:**

3: The reviewer is confident but not certain that the evaluation is correct

**Scope:**

3: The work is somewhat relevant to the Web and to the track, and is of narrow interest to a sub-community

---

### Official Review · Reviewer_f1RK · 2023-11-21

**Novelty:** 5
**Technical Quality:** 4

**Review:**

This study delves into the calibration of graph neural networks for semi-supervised learning through the manipulation of edge weights in the input graph. The authors introduce the concepts of "decisive" and "homophilic" edges, where the former denotes edges significantly influencing the trained neural network, and the latter signifies edges with endpoints bearing similar labels. In an initial experiment, the authors identified these edges using ground truth, empirically confirming that modifying them enhances neural network performance in terms of the expected calibration error (ECE). Subsequently, the authors propose heuristics for identifying these edges without relying on ground truth, and empirically validate that modifying such edges improves ECE across eight real networks.

While the methods employed are relatively straightforward, the resulting performance improvement is noteworthy. I commend the authors for the thoroughness of their experiments. However, a notable drawback is that the proposed calibration does not contribute to accuracy enhancement, which is the primary focus. Consequently, my feelings toward this paper are mixed.

**Questions:**

I don't have any question.

**Reviewer Confidence:**

3: The reviewer is confident but not certain that the evaluation is correct

**Scope:**

3: The work is somewhat relevant to the Web and to the track, and is of narrow interest to a sub-community

---

### Official Review · Reviewer_YHNE · 2023-11-23

**Novelty:** 4
**Technical Quality:** 5

**Review:**

Summary.   This paper addresses the challenge of miscalibration in Graph Neural Networks (GNNs), which limits the reliability of model decisions by affecting the models' awareness of prediction uncertainty. Traditional methods for improving calibration focus on modifying the GNN models themselves through techniques such as regularization or temperature scaling post-training. However, this study claims that miscalibration issues may be rooted in the graph data and that modifying the topology could be a solution. The authors observe that the GNN prediction is strongly affected by two types of edges within graphs: (1) decisive edges, which are crucial for GNN predictions, and (2) homophilic edges, which connect nodes of the same class. They find that weighting these edges more heavily in the adjacency matrix can enhance calibration performance without compromising classification accuracy. Based on these findings, the paper introduces a novel method called Data-centric Graph Calibration (DCGC). DCGC comprises two edge weighting modules: one that learns weights for decisive edges by parameterizing the adjacency matrix and allowing prediction loss to backpropagate to edge weights and another that assigns weights to homophilic edges based on predicted label distributions. In general, this paper successfully improves the calibration performance of  GNN prediction, despite some minor problems such as logical gaps and insufficient experiments.



Advantages. I would like to highlight the advantages of this paper below.
(1) This paper is generally well-organized and easy to follow. In particular, the thorough survey of current literature helps the audience better understand the technical contributions.
(2) The quality of most figures is high, and the meaning of every figure is clear. In addition, the organization of figures is also excellent.
(3) The research questions are properly answered with relevant technical contributions.

Disadvantages. I would like to mark out the disadvantages of this thesis below.
(1) The logic gap between experiment results and conclusions. For example, the authors’ motivation is derived from some calibration evaluation results. However, their evaluations are not sufficient to support their motivation.
(3) The lack of more extensive experiments. The paper only presents the calibration result of their proposed DCGC for two GNNs (GCN and GraphSAGE), which is not sufficient to prove the effectiveness of DCGC algorithm.
(4) The algorithm design needs more justifications. I am not convinced that either of them (edges with high gradients and hemophiliac edges) has anything to do with the calibration target, which tries to narrow the gap between average confidence and the average accuracy.

**Questions:**

Questions:

(1) The authors find that the Expected Calibration Errors (ECEs) on Cora (10.25%-18.02%) are always larger than those on Photo (4.38%-8.27%) across different graph neural networks (GNNs). Based on this observation, they try to calibrate the GNNs by assigning weights to the graph edges to calibrate the GNNs. However, the calibration performance difference on different graph datasets can be caused solely by graph node features but not graph structures. Did the authors try experimenting with different graphs with randomly generated node features to confirm their claim?
(2) This proposed pipeline is mysterious to me. The authors try to give edge weights such that the node classification prediction is more accurate and the neighbor nodes have a higher predicted class distribution. However, what is the relationship between theses and the calibration target, which tries to narrow the gap between average confidence and average accuracy?
(3) Are the graph adjacency masks $W_1$ and $W_2$ guaranteed symmetric? In the paper, the authors seem assigns two weights to one undirected edge, which is not interpretable in an undirected graph scenario.

**Reviewer Confidence:**

3: The reviewer is confident but not certain that the evaluation is correct

**Scope:**

3: The work is somewhat relevant to the Web and to the track, and is of narrow interest to a sub-community

---

### Official Review · Reviewer_oooW · 2023-11-24

**Novelty:** 6
**Technical Quality:** 5

**Review:**

Sumary: This study introduces a method for calibrating graph neural networks with emphases on data-centric approach. The method fine-tunes the input graph to enhance the alignment between confidence and accuracy. For node classification, this paper introduces two edge weighting modules, inspired by the observations of decisive and homophilic edges. The paper also shows the compatibility with existing temperature scaling-based methods. Furthermore, ablation study, efficiency and hyper-parameter sensitivity analysis is also provided.

Pros:
1. It innovatively proposes to calibrate GNNs from the DCAI perspective with model frozen, which is different from previous methods that focus on improving GNN model.
2. It proposes a novel calibration method named DCGC, which operates on the data level and can be easily combined with various temperature scaling-based methods.
3. It conducts extensive experiments and demonstrates that DCGC achieves SOTA calibration performance, while maintaining or even slightly improving accuracy.

Cons:
1. Decisive and homophilic edges are two heuristically observed factors. More comprehensive and automatic methods are needed in future work. And the usage of homophilic edges can be limited, i.e. It seems that  homophilic edges only apply to node classification problem.
2. Other than ECE, is there any other metric for calibration performance? Multi-dimensional metrics should be included for more comprehensive and fair comparisons.
3. The compatibility analysis for DCGC with in-processing methods need further investigation.

**Questions:**

Is there any other metric for calibration performance?

**Ethics Review Description:**

No ethic problem

**Reviewer Confidence:**

3: The reviewer is confident but not certain that the evaluation is correct

**Scope:**

4: The work is relevant to the Web and to the track, and is of broad interest to the community

---

### Official Review · Reviewer_jfc3 · 2023-11-29

**Novelty:** 5
**Technical Quality:** 5

**Review:**

Summary
This paper proposes a data-centric perspective to calibrate graph neural networks, which aims to modify the
graph structure for better performance in a general manner. Sufficient experiments are conducted against different baselines to show promising results. Overall the paper is well written with some minor issues.

Strengths:
- The flow of the paper is clear, the figures/diagrams used are helpful.
- Comprehensive experiments are conducted to show the effectiveness of the proposed method.

Weaknesses:
- The proposed method considers changing the graph structures by modifying the homophilic edge weights. This design choice needs in-depth justification. Is it possible to change graph structures by rewiring, or just removing less important edges?
- Only two base models are considered, which are GCN and GraphSAGE. It is unclear how general of the proposed method can be applied to more recent and more sophisticated GNN models.

**Questions:**

- Can the proposed homophilic edge weighting design be replaced with edge removal, or rewiring?
- How general can the proposed calibration be applied to more sophisticated GNN models?
- Can the observations discussed in this paper be applied to handle large-scale graphs?

**Reviewer Confidence:**

3: The reviewer is confident but not certain that the evaluation is correct

**Scope:**

4: The work is relevant to the Web and to the track, and is of broad interest to the community

---

### Decision · Program_Chairs · 2024-01-22

**Decision:**

Accept

**Comment:**

This paper presents an approach to improve calibration performance in GNNs by modifying the input graph structure. Their DCGC method adjusts the edge weights for decisive and homophilic edges of the input graph. The authors showcase improvements in expected calibration error over baseline GNN-specific calibration methods on 8 graph datasets without sacrificing accuracy.

 This was a borderline paper - while this is a relatively straightforward approach, the improvements in calibration performance for GCNs and GraphSage are promising. On the other hand, as some reviewers pointed out, the design choices seems quite heuristic-y and should be better explored and justified, either analytically or empirically. Furthermore, the paper would benefit from focusing its evaluation on newer GNN architectures (though the authors address this in their review response)